# Mechanisms underlying attraction to odors in walking *Drosophila*

**Liangyu Tao**[1], **Siddhi Ozarkar**[2], **Vikas Bhandawat**[1,2]*

**1** School of Biomedical Engineering, Sciences and Health Systems, Drexel University, Philadelphia, Pennsylvania, United States of America, **2** Department of Biology, Duke University, Durham, North Carolina, United States of America

* vb468@drexel.edu

## Abstract

Mechanisms that control movements range from navigational mechanisms, in which the animal employs directional cues to reach a specific destination, to search movements during which there are little or no environmental cues. Even though most real-world movements result from an interplay between these mechanisms, an experimental system and theoretical framework for the study of interplay of these mechanisms is not available. Here, we rectify this deficit. We create a new method to stimulate the olfactory system in *Drosophila* or fruit flies. As flies explore a circular arena, their olfactory receptor neuron (ORNs) are optogenetically activated within a central region making this region attractive to the flies without emitting any clear directional signals outside this central region. In the absence of ORN activation, the fly's locomotion can be described by a random walk model where a fly's movement is described by its speed and turn-rate (or kinematics). Upon optogenetic stimulation, the fly's behavior changes dramatically in two respects. First, there are large kinematic changes. Second, there are more turns at the border between light-zone and no-light-zone and these turns have an inward bias. Surprisingly, there is no increase in turn-rate, rather a large decrease in speed that makes it appear that the flies are turning at the border. Similarly, the inward bias of the turns is a result of the increase in turn angle. These two mechanisms entirely account for the change in a fly's locomotion. No complex mechanisms such as path-integration or a careful evaluation of gradients are necessary.

## Author summary

The strategy an animal employs to explore the environment and to find and return to the location where it has previously found food or mates is an important part of its behavior. In nature, animals have incomplete information about their environment, and must use this incomplete information to navigate. In most laboratory experiments, there is usually clear directional information making it difficult to infer an animal's real strategy from laboratory behavioral experiments. In this study, we devise a new behavioral task wherein we remotely activate olfactory neurons when fruit flies are in a given location. This activation makes a given location attractive to the flies without providing any directional information and allows us to assess how flies navigate under these conditions. We find that flies

**Data Availability Statement:** The data are held in a public repository (FigShare) https://doi.org/10.6084/m9.figshare.11356952.v1.

**Funding:** The work was supported by funding from the NIH RO1NS097881 to VB, from the NIH

RO1DC015827 to VB, and from the NSF IOS-1652647 to VB. The funders had no role in study design, data collection, and analysis, decision to publish, or preparation of the manuscript.

**Competing interests:** The authors have declared that no competing interests exist.

navigate towards the activated location using two simple mechanisms: First, its speed in the activated region and its turn rate is much lower than it is elsewhere. Second, at the boundary of the odor-zone, its speed decreases dramatically and its turns become much sharper. Essentially, these simple mechanisms appear to be extremely robust.

## Introduction

Movement is critical to an animal's survival. Much of the work on the neural mechanisms underlying the control of movement has focused on navigational movements. As a result, many strategies underlying navigation [1–3] have been uncovered. Even animals with relatively simple brains employ these strategies and can combine sensory cues, visual landmarks and path integration [4–6] to navigate towards a specific location. However, not every animal movement is aimed at reaching a specific destination and are therefore not navigational. During these movements–broadly described as search movements–the animal might have no or incomplete information about the resources it is seeking. Even expert navigators such as desert ants rely on local search once navigational mechanisms have taken them in the vicinity of their home [7]. Understanding an animal's search pattern in quantitative detail, and how it is altered by environmental stimuli or by navigational mechanisms as an animal becomes familiar with its environment is a fundamental question.

An obvious starting point for studying search movements is the study of movement in the absence of any information regarding the location of the resource that is being searched. This question has received much attention, particularly in the context of movement ecology [8,9]. The simplest approach to conceptualizing an animal's search movements in the absence of any other information is as a random walk where an animal walks straight in a given direction, and then changes its direction at random to walk in a different direction. The run-and-tumble model [10] used in bacterial chemotaxis is an example of a random walk. Random walk models also work well in some isolated cases of movements in larger animals [11]. However, the randomness in the change in direction makes this model quite limited because of limited directional persistence. Most animal movement is characterized by movement in the same direction for long periods of time, and therefore, exhibit greater directional persistence than can be modeled by random walk models. Two popular methods to model greater directional persistence are Levy walks [12, 13] and correlated random walks [14, 15]. Levy walks allow greater directional persistence by allowing a larger proportion of long-distance walks, while correlated random walks model persistence by allowing correlation in the direction of consecutive turns. Levy walks and correlated random walks are still "kinematic" or "non-orienting" models, i.e., the animal's movement in these models is not directed towards a particular destination by a navigational mechanism. Their success shows that non-orienting mechanisms can describe an animal's movement in a homogeneous environment. Although both models have had much success, Levy walks and correlated random walks are inappropriate as models in laboratory behavioral experiments in which animals typically spend a large fraction of their time exploring the boundary of the arena [16–21]. Moreover, in laboratory behavioral experiments, in which the temporal and spatial scales are short, stops can have a large effect on movement. Similarly, in a small arena, it is inaccurate to treat walks as straight walks (as in Levy walks) because even the small curvature of straight walk segments can have large effects on an animal's trajectory. Therefore, a new kinematic model is necessary to accommodate the behavior of animals in a small arena.

Kinematic models can describe an animal's movements in a homogenous environment because the animal does not have a strong directional preference. In some cases, a kinematic model is sufficient to describe an animal's behavior even in the presence of environmental cues. Bacterial chemotaxis is a prime example of a phenomenon that can be described by a kinematic model. Similarly, correlated random walks have been successful in describing the effects of stimuli [22] on behavior through non-orienting effects such as a change in speed or distribution of turn angles. For example, resource-rich habitats can produce lower speeds and more frequent, less correlated turns leading to an encamped walking pattern. In contrast, resource-poor habitats may result in explorative walks with higher speeds and correlated turns [23]. However, more generally as the animal becomes familiar with the environment or in the presence of directional environmental cues, in addition to kinematic changes, navigational and other orienting mechanisms can also play an important role. These mechanisms cannot be modeled by kinematic models. Indeed, even animals with simple brains such as *C. elegans* and *Drosophila* larvae respond to odors with strategies that are orienting [24, 25]. Adult *Drosophila* responding to odors show both non oriented changes in behavior [26,27], and are also capable of using orienting sensory cues such as wind [26, 28] to navigate to an odor source. Flies can also employ path integration to return to the location of food [29, 30]. Thus, a modeling framework in which the relative contribution of all of these mechanisms to locomotion can be studied is necessary.

In this study, we investigate a fly's locomotion in a small circular arena both in the absence of any stimulus and when its olfactory receptor neurons (ORNs) are optogenetically stimulated. The ORNs are activated only within a small central core of the arena. Immediately outside the central region, the light intensity decreases rapidly providing a directional cue that flies can employ to return to the center of the arena. Finally, there is an annular region within which the fly has little sensory information until the fly reaches the arena border. Thus, this arena allows an investigation of the interplay between orienting, non-orienting and path integration mechanisms in shaping a fly's movement. We also create an analytical framework in which these three mechanisms could be investigated within a single framework. We find that in the absence of ORN stimulation, a purely non-orienting mechanism is sufficient to explain the fly's distribution in our arena. When ORNs are optogenetically stimulated, a combination of orienting and non-orienting mechanisms are engaged to mediate a large change in the fly's distribution in the arena.

## Results

### ORN activation alone in the absence of directional wind cue can mediate robust attraction

Because ORN activation by itself does not carry directional information, activating ORNs is an ideal method for creating a stimulus without inherent directionality. However, in most studies of olfactory behavior (but see [31, 32] odors are delivered in a stream of air. The air stream is a strong directional cue; therefore, it is difficult to dissociate the effect of air on behavior from the effect of odor. To dissociate the two, we decided to optogenetically activate ORNs. Drawing inspiration from our earlier study [27] in which a circular region of uniform odor concentration was surrounded by a region in which there was no odor (See Methods for details), we created a similar arena but replaced odor with red light (627 nm). Because flies' photoreceptors have low sensitivity in the long wavelength, their behavioral response to red light, if any, is small. The arena was 8 cm in diameter; the central 2 cm region of the arena was illuminated with light of uniform intensity (light-zone). A sharp interface of 3 mm separated the light-zone from the rest of the arena (Fig 1A).

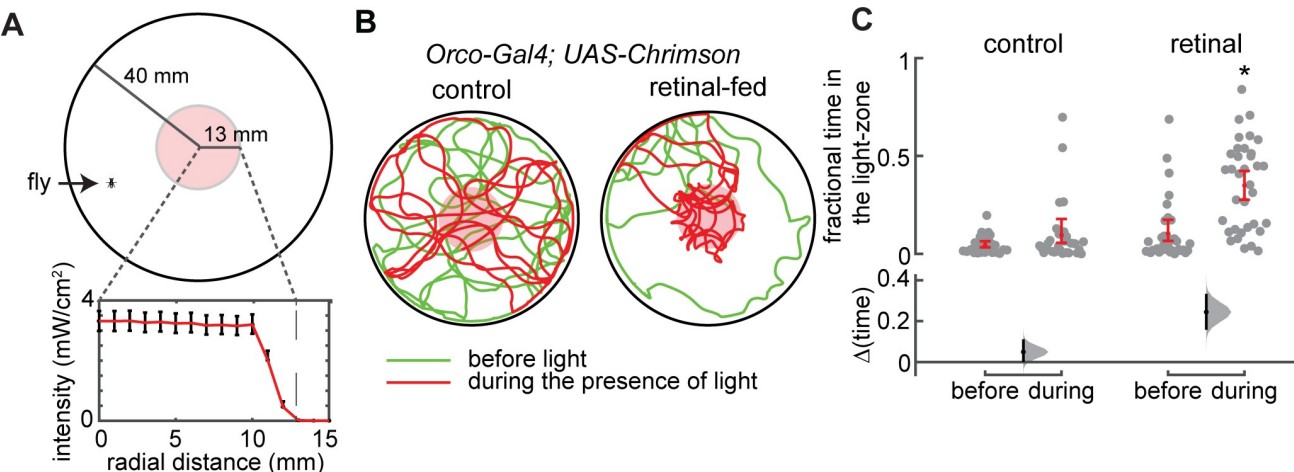

**Fig 1. ORN activation alone in the absence of directional wind cue can mediate attraction. A**. Schematic for the behavioral arena (for details see Methods and S11 Fig). The fly is constrained to walk in a 2D-plane. Light intensity as a function of radial distance shows a sharp border between 10 mm and 13 mm away from the center. Red line is the mean. Error bars represent the range of values. **B**. Sample trajectories showing that the retinal-fed flies spend more time inside the light-zone during the presence of light (red tracks) compared to the before period (green) and are, therefore, attracted to the light-zone and the control flies are not. **C**. Fractional time spent inside—a measure of attraction—for the Orco control (n = 31 flies) and Orco retinal flies (n = 35 flies) inside the light zone for the before and during periods. The fractional time spent inside for the Orco retinal flies for the during case is significantly higher than that in the before period (*, p < 0.01).

We used *Orco-Gal4* [33] to drive the expression of *UAS-Chrimson*, a red light-activated channel [34] in a large population of ORNs. Orco is one of the co-receptors for olfactory receptors and is necessary for activation of ORNs by odors [33], Orco-Gal4 expresses in about 70% of the ORNs. In the retinal-fed flies of this genotype (*Orco-Gal4; UAS-Chrimson)*, a large population of olfactory receptor neurons (ORNs) are activated when the fly enters the light-zone; flies on retinal-less food serve as the control because *Chrimson* needs retinal for activation.

Following the experimental paradigm that we have employed earlier [27], we recorded the fly's behavior three minutes before the light was turned on (green traces in Fig 1B, and in the three minutes during the presence of light (red traces in Fig 1B). The activation of *Orco*-ORNs dramatically changed the fly's behavior (Fig 1B). One change in the behavior is that the flies spend significantly more time inside the light-zone when the *Orco*-ORNs are activated than when the ORNs are not activated (Fig 1C). The fractional time a fly spends inside the light-zone is a measure of their attraction to the light-zone. The results in Fig 1 show that ORN activation alone in the absence of a directional wind cue is sufficient to elicit a strong behavioral response.

In the rest of the study, we will model these flies' locomotion and how their locomotion is modulated in response to odors. The main modeling goal was to develop the simplest model for a fly's distribution in the arena, and the mechanisms it employs to redistribute itself when the ORNs are activated. A fly's locomotion on a two-dimensional plane can be completely parameterized using three fly-centric variables—slip, thrust and yaw [35, 36] (S1A Fig). A recent study showed that a fly's locomotion is characterized by peaks in yaw which can be employed to understand the structure in a fly's behavior [36]. We reasoned that the same analytical framework can be used to model a fly's position, i.e., changes in yaw might be tightly linked to large changes in curvature and could anchor an analysis of a fly's movement. However, we found that peaks in yaw do not correspond in time to peaks in curvature. Furthermore, flies use slip, thrust and yaw flexibly to turn (S1 Fig and S1 Video) making it difficult to derive speed and curvature from slip, thrust and yaw. Therefore, to model how flies

redistribute themselves in the arena, we directly modeled a fly's locomotion based on its instantaneous speed and curvature.

### A fly changes its orientation using three mechanisms

An animal's track in a small arena cannot be classified *a priori* into straight walks and sharp turns because even the straight walks are curved. Moreover, the adult fly can turn sharply without stopping or reorienting itself following a sharp stop. Thus, qualitative observations of the data suggest that at least three states–walks, sharp turns and stops are necessary to describe a fly's locomotion. It is straightforward to identify stops as time points at which the fly's speed is close to zero (see Methods section on "Segmentation of tracks into boundary, walks, sharp turns and stops"). There are two possibilities regarding walks and sharp turns: If the curvature during walks and sharp turns lie along a continuum, then walks and sharp turns must be modeled as a single state. Alternatively, if the curvature during sharp turns is consistently higher than the curvature during walks, they would represent distinct states. To test whether curvature during sharp turns and walks are distinct, we segmented the tracks into putative sharp turns (regions with large changes in orientation), and putative curved walks (regions with little or gentle curvature). Sharp turns were identified by using a custom algorithm (see S2 Fig and Methods for the algorithm); regions between consecutive sharp turns were curved walk. We compared the distribution of average curvature of curved walk segments and sharp turn segments. Based on receiver operating characteristics (ROC) analysis on these distributions, sharp turns and curved walks were assigned accurately with 97% confidence using a binary logistic regression classifier based on the average curvature (S2 Fig). This implied that sharp turns and curved walks can indeed be segmented into two separate states; and overall a fly's tracks could be segmented into three states–sharp turns, curved walks and stops.

Because flies tend to move over relatively long periods with similar speed and curvature (37), it is likely that each transition to one of the three states can be defined by a few parameters: Modeling stops is straightforward; stops can be modeled by two parameters—stop duration, and the change in a fly's orientation during the stop. To investigate whether curved walk and sharp turns can be modeled using a few parameters we compared empirical track segments to synthetic ones that represented abstractions of the empirical tracks. A fly's speed during both sharp turns and curved walks could be modeled by its mean speed. The change in orientation during a sharp turn can be modeled as an instantaneous change in curvature equal to the sum of curvature at the point of the local peak in curvature (Fig 2A$_1$). The change in orientation during curved walks can be modeled by assuming that a fly has a fixed curvature throughout a given curved walk (Fig 2A$_2$). Examples in Fig 2A show that both curved walk and sharp turns can be modeled using three parameters.

Given that we could accurately model the change in the fly's orientation during a given segment, we checked whether a series of stops, sharp turns and curved walk can model the overall reorientation of a fly's track during the entire experiment. To this end, we computed the cumulative sum of the curvature for the trajectory of each fly and compared this sum to the cumulative sum resulting from the model of sharp turn, curved walk and stops. To assess the relative contributions of sharp turn, curved walk and reorientation during stopping, we individually removed each type of reorientation and compared the resulting cumulative sum of the curvature to that of the empirical data (Fig 2C) and found that the RMSE was significantly higher when any of the three types of locomotion were removed (Fig 2C). In sum, a fly's movement can be described in terms of three states–stop, curved walk and sharp turns each of which can be described by two to three parameters.

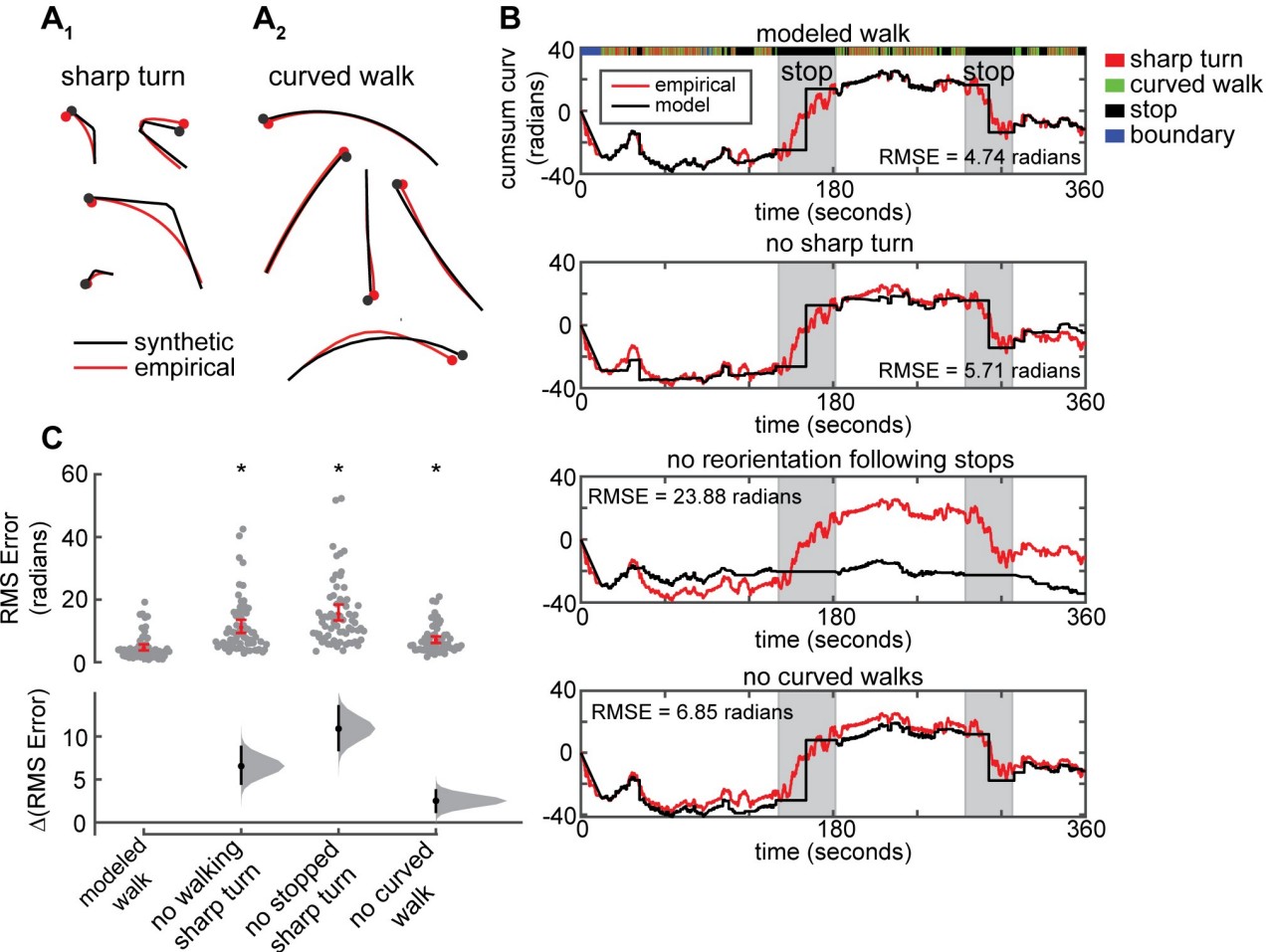

**Fig 2. Three ways to change orientation: curved walk, sharp turn and stop. A.** Synthetic tracks generated using mean speed and curvature closely approximate empirical tracks. Examples of sharp turn (left) and curved walk (right) are shown. Filled circle marks the end of the track. **B.** Cumulative curvature of the empirical track is compared to the modeled track, and to the modeled track without one of the three states. **C.** The distribution of mean RMSE of the cumulative sum of the curvature for each model for all flies (n = 66). Taking out any of the three forms of reorientation significantly deteriorates the ability of the model to approximate empirical change in curvature (*, p < 0.01).

## A four-state kinematic model can model a fly's locomotion but not its response to odors

The three kinematic states–stop, walk and sharp turn describe a fly's behavior everywhere in the arena except the boundary; the fly's behavior at the boundary is fundamentally different from its behavior away from the boundary because the fly usually circles the boundary, and the fly's behavior at the boundary is well described by two parameters–the time the fly spends and its angular velocity at the boundary. To examine whether these four states (Fig 3A) can model a fly's position in the arena, we generated synthetic tracks. Just like the experimental flies, each synthetic fly walked for 6 minutes—3 minutes before the light turned on, and three minutes following light on. Synthetic tracks were composed of sequences of the four states. Synthetic flies started at the center of the arena and moved around the arena through a succession of transitions into the four states. Synthetic flies always started in curved walk. Curved walk ended in stop, sharp turn or at the boundary (Fig 3A). Each of these three states ended in a new curved walk. Tracks corresponding to each transition were generated as described in Fig

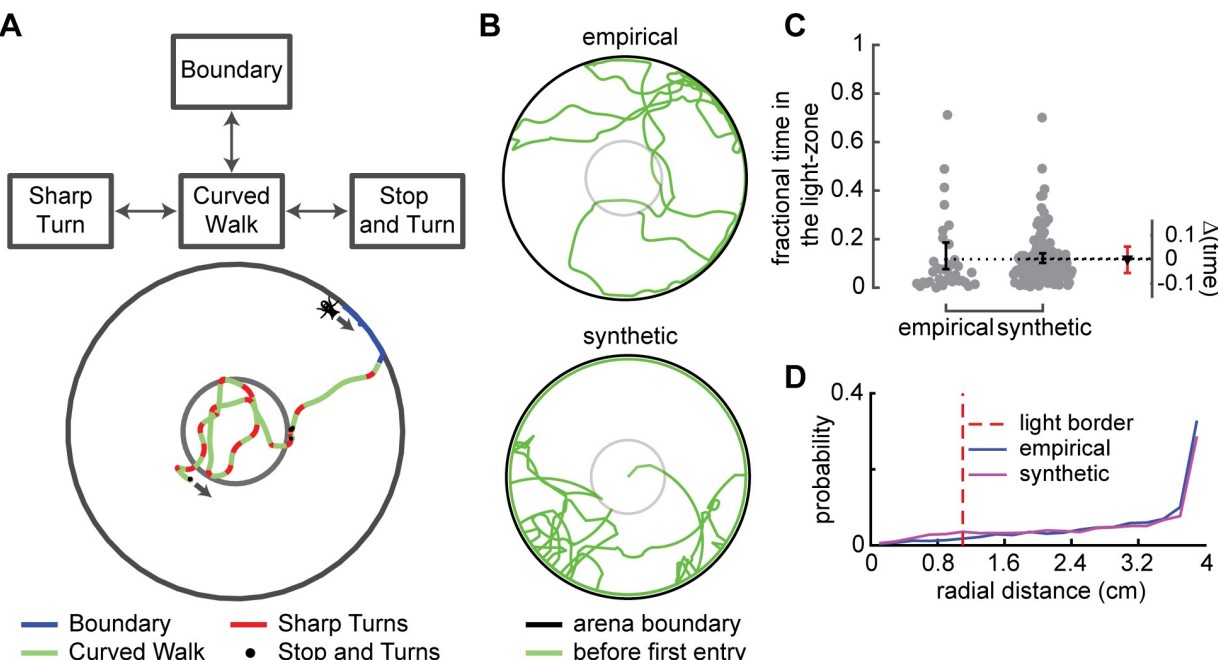

**Fig 3. The walk-turn-stop-boundary (WTSB) model can describe a fly's distribution in the arena in the absence of ORN activation. A.** Curved walk ends either when the fly reaches the arena boundary, makes a sharp turn or stops. **B.** Example empirical (top) and synthetic (bottom) tracks before the light is turned on. **C.** The time that synthetic flies spend inside the light-zone is not different from that for the empirical flies. **D.** The radial occupancy for the synthetic flies is similar to the empirical flies.

2 (see Methods for details) by sampling from speed, curvature and duration distributions for each state (S3 Fig and S4 Fig). Consistent with previous work [27,37], the behavior of the fly is different inside the light-zone and outside it, therefore three different distributions were used to model the fly's behavior–before the presence of light, during-inside and during-outside. The duration that each transition lasted was also selected from the empirical distribution (S3 Fig and S4 Fig). Because our experimental flies were selected to enter the light-zone at least once both in the before and during period, the synthetic flies also went through the same selection criteria (see Methods for details). In all, 116 flies fulfilled our selection criteria; one example is shown in Fig 3B, and the entire set is shown in S7 Fig. The algorithm for generating synthetic tracks is detailed in S5 Fig. The behavior of synthetic flies depended solely on speed and curvature. Therefore, the model described in Fig 3 is a kinematic model or a non-orienting model. Moving forward, we will refer to this model as the Walk-Turn-Stop-Boundary (WTSB) model.

To test whether the flies generated using the WTSB model resemble empirical flies in their distribution in the arena we compared the distribution of times a fly spent within the light-zone. The distributions of the empirical and synthetic flies are similar (Fig 3C). In another, perhaps more nuanced measure, we compared the radial distribution of empirical and synthetic flies and found that the two radial distributions were quite similar (Fig 3D). Therefore, the kinematic model is an adequate model of a fly's distribution in the arena before optogenetic activation of ORNs.

Given that the model describes a fly's locomotion before activation well, we investigated whether the model can also describe modulation of locomotion upon ORN activation. ORN activation modulates many aspects of a fly's locomotion (S3 Fig and S4 Fig). The changes in a

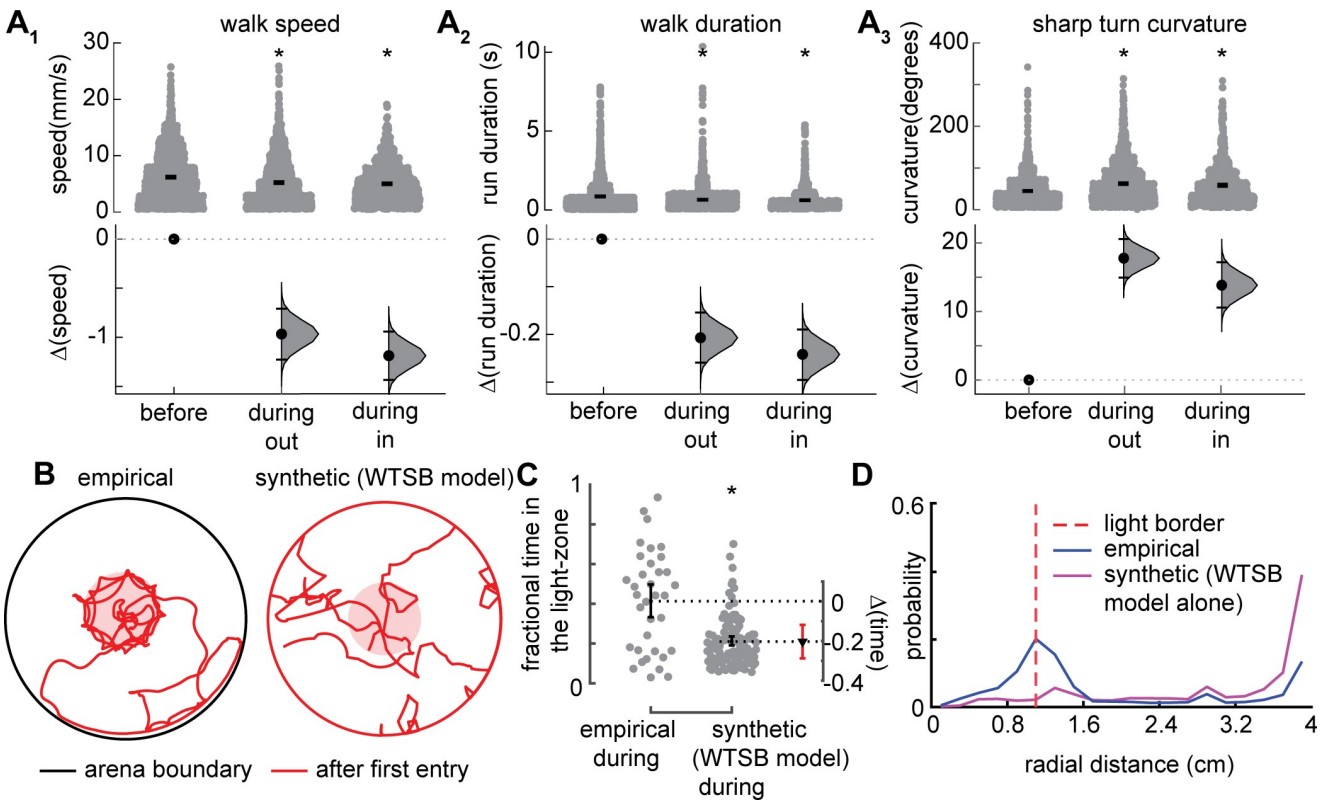

**Fig 4. ORN activation affects many aspects of a fly's kinematics, but kinematic changes do not explain the increased time a fly spends inside the light.** **A.** Kinematic parameters modulated in the Orco retinal flies: There is a large decrease in walking speed (A1) and duration (A2), and increase in curvature of sharp turn (A3) (*, p < 0.01). **B-D.** Synthetic fly track generated using the WTSB model based on the changed kinematic parameters do not show the same attraction as the empirical flies. **B.** Example flies. Pink region designates light region. **C.** Experimental flies are significantly more attracted than the synthetic flies generated using the altered kinematics incorporated using the WTSB model. **D.** Radial occupancy for the empirical and synthetic flies for the period after first entry. The dotted red line shows the light border.

fly's walking speed, duration of its walks and the sharpness of its turns (Fig 4A$_{1-3}$) were statistically significant.

To assess whether the changes depicted in Fig 4A can explain the change in distribution following optogenetic activation, we generated synthetic flies with the altered kinematic parameters using the WTSB model. We compared the spatial distribution of synthetic flies in the presence of ORN activation (S7 Fig, red tracks) to that of the empirical flies. A visual inspection of the tracks (Fig 4B and S6 Fig) suggests that the synthetic flies are not as attracted to the light-zone as the empirical flies. Indeed, the time that the synthetic flies spent inside the light-zone is significantly less than the time spent by the empirical flies (Fig 4C). The radial distribution is also markedly different (Fig 4D). Therefore, the large kinematic changes when the ORNs are activated is not sufficient to mediate the large attraction observed when the ORNs are activated. However, the kinematic changes do mediate a small, but significant attraction (S9 Fig).

## Increased and directed turning are necessary for a fly's attraction to odors

That the WTSB model describes locomotion in the absence of ORN activation but not in its presence is likely because WTSB is a non-orienting model. A non-orienting model is sufficient in a homogenous environment but fails when the environment is no longer homogeneous.

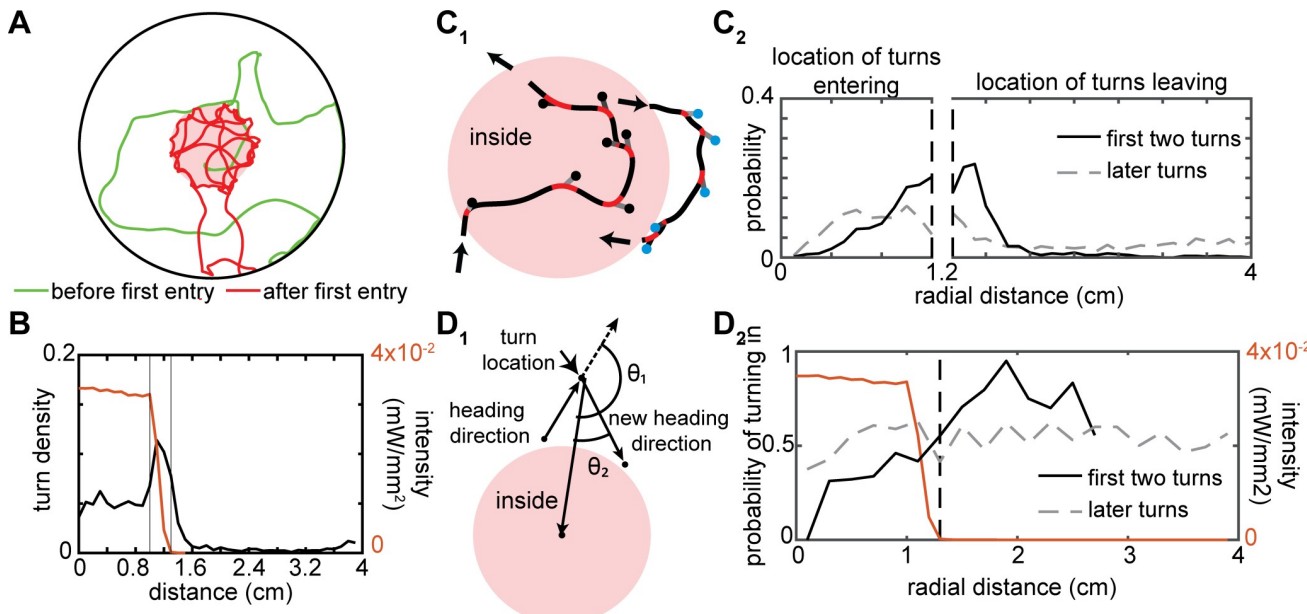

**Fig 5. Flies turn more at the border and these turns are biased such that flies trajectory is directed towards the center of the arena. A.** Sample fly track shows increased turning near the light edge (pink region). **B.** Flies exhibit an increase in the number of turns within the odor boundary. This increase coincides with decrease in light intensity (in red). Thin black lines mark the region where the light intensity decreases from full intensity to no light. **C.** Increased tendency to turn at the light-border. $C_1$. Schematic showing how the graph in $C_2$ was generated. The fly's trajectory is split into outside and inside trajectories based on the head position. Sharp turns are indicated by red segments and the location of sharp turn is indicated with blue points (outside) and black points (inside). Gray lines represent the body axis. $C_2$. The probability of making the first two turns increases near the light-border at (1.3cm). The probability of remaining turns is about the same. $D_1$. Definition of turn bias. The turn is inward when $\theta 1 > \theta 2$. $D_2$. The first two turns (shown by the solid line) made by the fly as it enters or leaves the odor zone show a directional bias so as to keep the fly inside the odor zone. This bias disappears during the later turns (shown by the dotted grey line). The red line shows the spatial profile of the light intensity.

The flies' tracks suggest a directed element (Fig 5A) when ORNs are active: Many flies appear to weave in and out of the light-zone (Fig 5A and S6 Fig). Indeed, there is a large increase in the density of sharp turns at the border of the light-zone (Fig 5B). Therefore, it appears that flies can sense the decrease in light intensity as they exit the light-zone and turn to re-enter the light-zone.

As a first step towards modeling this orienting response, we measured the increase in the fly's tendency to turn as it crossed in or out of the interface between the light-zone and the no light-zone (Fig 5C$_1$), and found that the increased turn density is almost entirely due to the first two turns after the fly crosses the interface (Fig 5C$_2$). The increased tendency to turn was modeled by introducing a "border choice parameter". Importantly, a large fraction of these turns redirect the fly towards the light-zone because the angle between the radial vector and the heading direction after the turn is smaller than the corresponding angle before the turn (Fig 5D$_1$ for definition, and Fig 5D$_2$ for empirical data). Thus, the turns are biased, and to model this bias we introduced the "turn-bias parameter" that quantifies the fraction of turns that are inwardly directed. Both the "border choice" and "turn-bias" only affect the first two turns after the fly exits the arena (Fig 5C and 5D).

To investigate the effect of this directed response on attraction to the light-zone, we modified the kinematic model to include border choice (BC) and turn-bias (TB) parameters (n = 171). All the features of the model in Fig 3 are preserved; in addition, the fly makes more turns at the interface, and these turns are biased (see Methods). We refer to this model as the WTSB+BC+TB model. One example of synthetic flies (using the WTSB+BC+TB model) is shown in Fig 6A. The entire set of synthetic flies can be found in S8 Fig. The synthetic flies

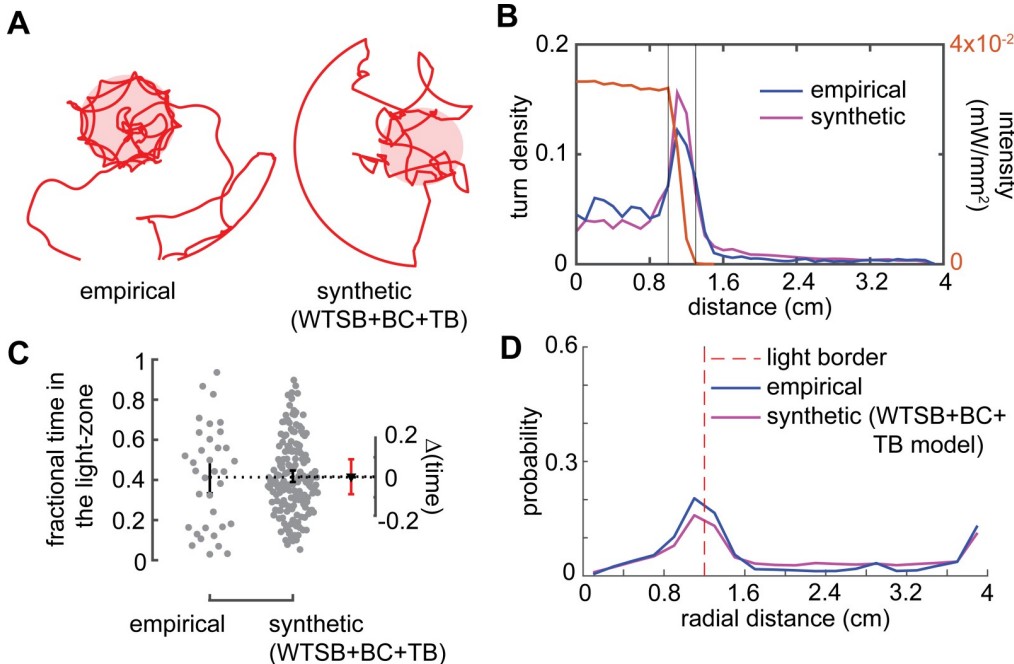

**Fig 6. A model that incorporates border choice and turn bias can describe attraction. A.** Sample empirical and synthetic tracks (using WTSB model + border choice(BC) + turn bias (TB)) after first entry into the light-zone. Pink region designates light ring. **B.** The synthetic flies show the same turn density as do the empirical flies. **C.** Attraction index for the synthetic flies after first entry is not different from that of the empirical flies. **D.** Radial occupancy for the empirical and synthetic for the period after first entry. The dotted red line shows the light border.

have a similar propensity to turn at the border as do the empirical flies (Fig 6B). The synthetic and empirical flies spend similar time inside the light-zone (Fig 6C). Furthermore, the radial density of the synthetic flies matched that of the empirical flies (Fig 6D) implying that with the addition of the orienting response, the change in the flies' distribution could be accurately modeled.

## Mechanisms underlying orientation response: Slow down and turn hard

The orienting response observed in Fig 5 could result from either sensory cues or path-integration or a combination of the two. There are two components of the directed response: increased turning at the border of the light-zone and turn-bias. First, we will consider the mechanism underlying increased turning (Fig 7A and 7B), and then the mechanism underlying turn bias (Fig 7C).

One possible mechanism underlying increased turning at the border of the light-zone is osmotropotaxis–a process by which the fly compares the stimulus intensity incident on the two antennae and turns towards the antennae that receives the stronger stimulus [38, 39]. To evaluate whether osmotropotaxis is the dominant mechanism underlying directed turns, we removed the olfactory organs (antennae) on the right-side and examined the behavior of flies (Fig 7A$_1$, tracks from all the flies are shown in panel C of S10 Fig). Unilateral antennae removal did not induce turns in one direction; there was no difference between the number of turns towards the left versus the right. Flies with the olfactory organ on one-side spent a similar time inside the light-zone as did the empirical flies (Fig 7A$_2$). The experimental flies also had a similar spatial distribution (Fig 7A$_3$) upon ORN activation as did the control flies. Thus, it seems unlikely that the comparison across the two antennae is the dominant mechanism underlying

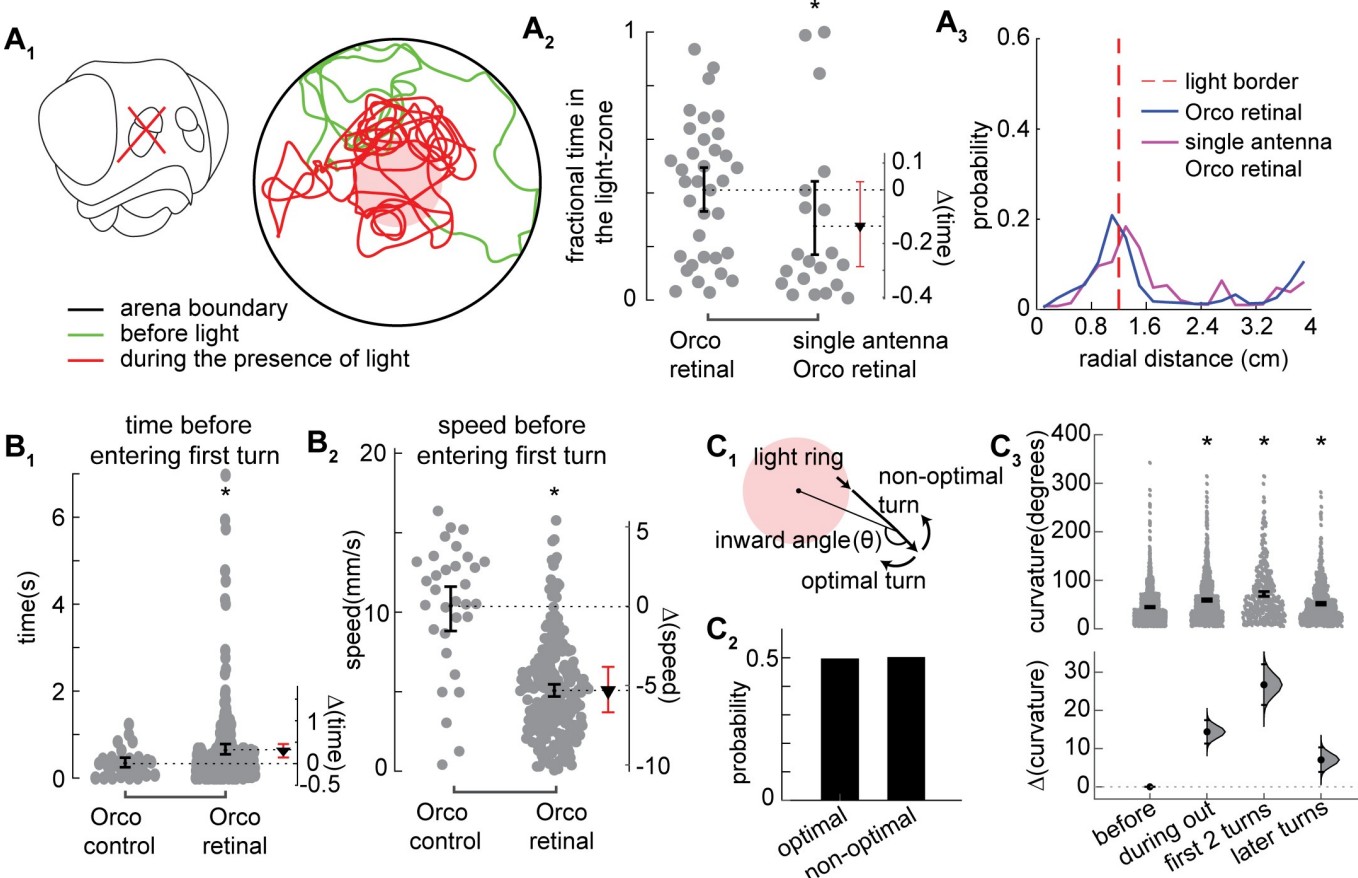

**Fig 7. Increased inward turn is a result of slowing down and turning hard at the light border. A.** ORNs are located in the antennae. Experiments with the right antennae removed causes unilateral sensing. **A₁.** Schematic and example tracks. **A₂.** Single antenna flies show a lower attraction as compared to Orco retinal flies (but not statistically significant) and in **A₃** show a comparable distribution of radial occupancy (*, p < 0.01). **B.** Although the flies turn after a similar time in the presence of light (**B₁**), because they move slowly, these turns are closer to the border (**B₂**). **C₁.** Schematic showing optimal and non-optimal turn. **C₂.** Flies have the same chance of executing optimal and non-optimal turns as they leave the light ring. **C₃.** Orco flies significantly increase their sharp turn curvature at the odor-border leading to large turn-bias.

the orienting response. Although not central to this study, it is important to note for completeness that the left turn inside the light-zone were much wider than the right turns (S10 Fig).

An alternate mechanism could be temporal comparison or klinotaxis [40, 41]. As a fly exits the light-zone, the decreasing light intensity results in a decrease in the ORN firing rate, this decrease in ORN firing rate could result in turns immediately following exit from the light-zone. If decrease in ORN firing rate were the mechanism that induces sharp turns, the time elapsed before the fly makes its first sharp turn after it exits the arena should be shorter in the presence of light. Surprisingly, the time elapsed before sharp turn is similar in both the retinal and control flies (Fig 7B₁). The dominant reason for the increased density of turn is a large decrease in speed as the fly exits the arena (Fig 7B₂). Because the fly is traveling slowly before it makes its first turn, the turns are closer to the arena border. The increased turning at the border is due to the decrease in speed as the fly exits the light-zone.

A possible mechanism for turn bias is an idiothetic mechanism as has been suggested by others [29, 30]. As the fly exits the light-zone, unless the fly is walking along the radial vector, it can either turn in the optimal direction to return to the odor-zone or in the non-optimal direction (Fig 7C₁ for schematic). A higher percentage of turns in the optimal direction would

support the idea that an idiothetic mechanism is being employed. However, we found that the probability of turns in the optimal direction is the same as the turns in the non-optimal direction (Fig 7$C_2$) ruling out an idiothetic mechanism. Instead, the most likely mechanism for the large turn bias is the fact that the angle through which a fly turns during each sharp turn is much larger (Fig 7$C_3$). The large turn angle means that many turns in the non-optimal directions also tend to reorient the flies inward.

## Discussion

The success of the WTSB model in describing a fly's locomotion in the absence of ORN activation provides two important insights. First, unsurprisingly, a non-orienting strategy is enough to describe a fly's locomotion in the absence of ORN activation. In a small, dark arena, there is little to orient a fly's locomotion in a given direction. Instead, the fly searches the arena without any specific navigational goal. The fly spends large chunks of its time at the arena border. On an average, each excursion to the arena border lasts 7 seconds. Inside the arena, the fly's behavior is well-described by three states–walks, sharp turns and stops. Each of these states are kinematic states or non-orienting states because the behavior of the fly within each state can be modeled by sampling from a speed-curvature distribution. Second, the behavior within the curved walk is more sophisticated than many kinematic models employed (29) to describe them which model curved walks as straight walks with constant speeds; only the duration of the walks is varied. We found that each curved walk is described by speed, curvature and duration implying that flies not only select the duration of each walk, but also its speed and curvature.

It is noteworthy that each state is remarkably well-described by the mean speed and curvature. Since each state lasts 1 second on an average, it implies that the fly makes one decision (in selecting speed and curvature) every second on average. This feature of the model is consistent with the Hierarchical Hidden Markov Model (HHMM) employed in a previous study [37]. HHMM showed that the fly moved at similar speed and curvature over long durations. This consistency is likely why the average speed and curvature is an accurate descriptor of a fly's kinematics within a state. This consistency is not just a matter of detail; it reflects a fundamental feature of search movements because it implies that the fly's behavior unfolds in chunks of tens of steps during which the fly uses the same speed and curvature. This chunking implies that flies would make fewer decisions per unit time than implied in continuous-time models employed by others [26]. Because the flies are making fewer decisions per unit time, there will be greater variability between different instances of tracks generated from the same underlying model, and this contributes to the observed inter-individual variability [37].

Consistent with our previous work [27, 37], activating ORNs affects a fly's behavior not only inside the light-zone (or odor-zone in the previous study) but also outside the light-zone where there is no light and the ORNs are presumably not active. Thus, ORN activation influences behavior both through direct sensory-motor transformations near the light-zone, and indirectly through mechanisms that likely rely on memory. Many of the changes we observe are "kinematic" changes or non-orientational changes. These changes–such as decrease in speed, run duration and increase in turning in the presence of odor–are consistent with changes observed in field behavior in insects [42], as well as consistent with previous work [27] in our lab using odor stimulation and not optogenetics. In a previous study, we found that a strong attractant–apple cider vinegar–also caused a decrease in speed inside the odor-zone. This decrease is consistent with recent work in flying *Drosophila* that analyzed changes in a fly's behavior when odors were presented in the context of visual objects [31]. In other experiments, there is an increase in speed upon odor encounter[26, 43–45]. An elegant explanation

for this difference is that flies slow down if they expect to encounter an odor object, and move faster if they expect the object to be far away[31]. However, it is important to make two observations regarding these non-orientational changes. First, as we have shown in a previous work, different attractive odors produce different kinematic changes [27]. Second, that, at least in the present experiments, they only make a small contribution to the redistribution of flies in the arena. Much of the attraction is mediated by the fly's increased propensity to turn at the border between the light-zone and no light-zone. Because most studies of insect olfactory behavior focus on orientational mechanisms, particularly orientation to wind, the importance of non-orientational mechanism in different behavioral contexts is poorly understood.

At first glance, the orientation behavior that we observe here appears similar to the orientation behavior reported in many other insects both during walking and during flight. Almost all insects exhibit the ability to turn back into the region of high odor concentration. However, the mechanism they employ appears to be diverse. One mechanism suggested by Kennedy for odor-tracking in moths [46], and subsequently also considered to be important in other insects [47, 48], is the automatic internally stored counter-turning which is released when the odor concentration decreases and brings the insect back into the odor plume. Another mechanism, in contrast to counter-turning, is a direct increase in turning in response to decrease in odor concentration [26]. Both mechanisms depend on increased turn rate. In contrast, the mechanism we demonstrate here appears to be novel because of the absence of increased turning. Rather, the sharp decrease in speed makes it appear that the flies turn more as they exit the light-zone. Another unexpected result is that the angle through which the fly turns at the light-no-light interface is much larger than the turn angle at other locations. Aspects of the mechanism that we demonstrate here-slow down and turn hard—have been observed in other cases. In many studies (for example Bell and Tobin 1981 [21]) the rate of turning per unit distance was shown to increase, a result similar to our observations. Similarly, there is also a precedence for the increase in turn angle [49].

The slow down and turn hard mechanism is simpler to implement than other mechanisms that have been proposed for the orienting behavior such as osmotropoataxis. It is certainly simpler than mechanisms based on spatial memory or path-integration. Essentially, a circuit that relates decreasing ORN response to decreased walking speed and increased turn angle during sharp turn is all that is necessary to produce robust attraction to a stimulus. This strategy is also generalizable beyond active ORN or odor as the stimulus; essentially, any sensor that can detect a gradient can be combined with a slowdown-and-turn-hard module to affect robust attraction. The turn-bias we observe in this study is strikingly similar to the results in a previous study by Kim and Dickinson [29]. As in this study, in the Kim and Dickinson study, flies increasingly return to a tastant by biasing their turns. While in the Kim and Dickinson study, the authors interpret this turn bias as evidence for path integration, we demonstrate that a similar result can be obtained simply by slowing down and turning more sharply.

Given the elegant demonstration that *Drosophila* projection neurons respond differentially to stimulation of ipsilateral versus contralateral antennae [39] and demonstrated osmotropotaxis in multiple insects [38, 50–52], it is somewhat surprising that a bilateral comparison was not a dominant mechanism in the experiments performed in this study because the sharp light gradient makes the experimental condition ideal for osmotropotaxis. One possible explanation for the lack of a prominent role of osmotropotaxis is that the distance between the two antennae is not large enough, and therefore even with a steep light gradient the light intensity across the two antennae are not different enough to be detected over neural and environmental noise. Another possibility is that the dominant mechanism underlying the behavior in this arena is inhibition of ORNs as the fly moves from a region of high light intensity to low light intensity. Once an ORN's firing rate reaches zero, differences in odor concentration between

the two antennae would be difficult to detect. This interpretation is supported by the finding that the first two turns inside the light-zone display a left-right asymmetry whereas the first two turns outside do not. An osmotropotactic mechanism might be more useful in following a linear trail where the stimulus gradient is normal to the inter-antennal axis. Finally, that the bilateral input is not required is not particularly surprising because similar results were observed for fly larvae [51] and ultimately might reflect the fact that redundant mechanisms are at play; another study examining the algorithms underlying odor tracking in wind also reached a similar conclusion[26].

Strategies that animals employ to control their movement range from pure search movement in the absence of information about the environment to purely navigational movements. The WTSB model provides a flexible framework for investigating locomotion across a range of these strategies. Some insights that we obtain here are strikingly similar to the insights that we obtained using the HHMM model that we employed to model a similar dataset [27, 37]. The HHMM model we employed used variables similar to speed and curvature and returned states that had characteristic speed and curvature. Therefore, the HHMM that we obtained functioned as a kinematic model. The advantage of the WTSB framework is that it provides a flexible framework that can be tuned by the experimenter to evaluate different mechanisms underlying a fly's behavior. Indeed, when we applied border choice and turn-bias (WTSB +BC +TB model in Fig 6), we were able to replicate the distribution of the fly in the arena. This ability to tune is likely a disadvantage when one is trying to establish differences between flies in which case a more rigid model such as an HHMM is likely a better choice.

## Materials and methods

### Experimental model and subject details

Drosophila melanogaster strains were raised in sparse culture condition as described in an earlier study [43]. 100–150 eggs were collected in standard cornmeal media and cultured in the incubator at 25°C in a 12hr dark/12 hour light cycle. All experiments were performed on adult female *Drosophila*. Newly eclosed adult female flies were transferred to fresh food vials wrapped in aluminum foil to prevent exposure to light. Half of the progeny were transferred to food without all trans-retinal (control flies), and the other half were transferred to food containing all trans-retinal (retinal flies). 10–15 flies were starved in a foil wrapped empty vial with a wet Kimwipe for 15–21 hours prior to the experiments. Experiments were conducted on control flies 3–5 days after eclosion, while experiments on retinal flies were conducted 4–5 days after putting them on the retinal food vials to allow more time for retinal to incorporate into *Chrimson*. All experiments were performed using *Orco-Gal4*;*UAS-Chrimson* flies.

### Behavioral arena and experimental setup

The behavioral arena had a design similar to the arena used in a previous study[27]: The arena was circular with a radius of 40 mm, and the flies were constrained to walk because they were between two plexiglass plates. A central region of radius 12.5 mm could either be illuminated with a red light or it could be without a red light. A fly walking into the center when the light is turned on would have its ORN activated and would therefore "smell" an odor.

The arena consisted of the following parts placed on top of the each other– 1) Ø90 mm diameter and 3 mm high black Delrin plate with a Ø25 mm concentric hole for the red light was the bottom-most piece, 2) On top of the black Delrin piece were two Ø90 mm x 1.5 mm plexiglass plate; a 3 mm tall, Ø90 mm outside diameter and Ø80 inside diameter Delrin ring served as a spacer (S11 Fig). Flies walked between the two plexiglass plates.

Light was delivered using a red (617 nm) light emitting diode (LED) (Thorlabs M617L3) directly beneath the central light ring. The LED light was collimated (Thorlabs ACL2520U) and focused using a plano-convex lens (Thorlabs LA1433). Light intensity was measured using a photometer (Thorlabs S121C) at increments of 1 mm radial distance away from the center of the arena when the light was turned on. Fig 1A shows the mean, minimum, and maximum intensity over 10 measurements at each radial distance. The arena was lit with two infrared (850 nm) light sources (Lorex vq2121). Videos were acquired at 30 frames per second using a machine vision camera (Marlin 131B Camera, Allied Vision Technologies with FUJINON 1:1.4/9mm lens).

Flies were given a 5-minute light acclimation period followed by a 10-minute dark acclimation period that reflected experimental conditions.

## Tracking the fly and assigning its head position

A custom-made MATLAB program was written using the MATLAB Image Processing Toolbox (Mathworks, Natick, MA Release 2018a) to automatically track the fly. Briefly, for any given video, a circular region of interest outlining the arena boundary was selected by the user. All subsequent processing steps were limited to this circular region. An average background frame was calculated for both the before light-on and during light-on periods by averaging randomly selected frames. The respective backgrounds were subtracted from each frame. In the background-subtracted frame, the fly could then be detected as an ellipse using the Matlab function *regionprops.m*. We defined the fly as the largest ellipse with a minimal area of 1.57 mm$^2$ (1 mm semi-major axis x 0.5 mm semi-minor axis ellipse).

The major axis of the ellipse represented the long axis of the fly; the centroid of the ellipse represented the centroid of the fly. For each ellipse, we obtain tentative head and tail locations from the major axis. Starting from the second frame, we applied the Hungarian algorithm to reassign the head and tail to the tentative locations by minimizing the total distance traveled by the head and tail from the previous frame. Head position assignments were then corrected using two post-processing steps: 1) Flies do not move backwards for extended periods of time. 2) Flies do not turn at an angular speed > 120 degrees /frame (3600 degrees/second). The position of the fly on frames with no ellipses that fulfill this criterium (<1% of the total number of frames) were linearly interpolated based on the fly's position on the frame before and after the frame under consideration. Finally, raw trajectories of the centroid and orientation were smoothed utilizing a 1 and 1.3 second LOESS filter respectively. Regions where the error in smoothing resulted in a larger than 0.0037 mm difference from the raw values were instead smoothed with a 0.2 and 0.3 second LOESS filter respectively.

Following the post-processing steps, we validated a sample of 400 frames across 10 videos by manually labeling the head and body positions. The median error was 0.15 mm for the centroid, 0.16 mm for head position, and 3.53 degrees for the orientation. The error was higher at the arena boundary compared to the rest of the arena.

**Table 1. Weights used to delineate sharp turns from curved walks.**

| Description | Value |
|---|---|
| Curvature threshold | 13.843 degrees/frame |
| Change in curvature threshold | 4.080 degrees/frame$^2$ |
| Curvature weight | 0.890 |
| Change in curvature weight | 0.739 |
| Sharp turn threshold multiple | 1.365 |
| Curved walk threshold multiple | 0.230 |

## Calculation of kinematic parameters from tracks

From the fly's position and orientation, we obtained slip, thrust and yaw. We also obtained the speed and curvature of its track. These calculations are described below.

First, we describe the calculation of slip, thrust and yaw. Given the two consecutive center of mass positions ($p_1,p_2$) and the orientation of the drosophila body ($\theta$), we can define the movement angle between the direction of movement and the body orientation as

$$\varphi = \theta - \tan^{-1}\left(\frac{<p_2 - p_1>\hat{j}}{<p_2 - p_1>\hat{i}}\right)$$

And the speed (s) as

$$s = \frac{p_2 - p_1}{\Delta t}$$

Defining thrust (T) as the movement of the fly along the main body axis and slip (S) as the movement of the fly perpendicular to the movement axis, we obtain:

$$T = s \cdot \cos(\varphi)$$

$$S = s \cdot \sin(\varphi)$$

Finally, we can define yaw as $Y = \theta_2 - \theta_1$.

Next, we calculated curvature as follows: At any position, the direction of the movement trajectory can be approximated by the change in position one step prior and one step after the position, normalized by the speed. The normal vector ($N$) is defined as the vector normal to this movement trajectory and can be calculated as follows:

$$\alpha'(t) = -\left(\frac{dy_t}{s_t} + \frac{dy_{t+1}}{s_{t+1}}\right)\hat{i} + \left(\frac{dx_t}{s_t} + \frac{dx_{t+1}}{s_{t+1}}\right)\hat{j}$$

$$N = \frac{\alpha'(t)}{|\alpha'(t)|}$$

The curvature is defined by the change in the angle of the normal vector: $k = dN$.

## Segmentation of tracks into the four states: Boundary, walk, sharp turns and stop

Assigning the fly as part of boundary or stop state was straightforward:

Boundary: When the centroid was within 1.5 mm (half a fly length) of the arena boundary, the fly was assigned as being in the boundary state.

Stop: When the fly's speed was less than 0.5 mm/s, we assigned the fly as being in the stop state. The rest of walking tracks were further delineated into sharp turns and curved walks as follows.

We first binarized the tracks using a threshold (see Table 1) for the absolute value of the curvature and a threshold for the derivative of the curvature separately. We then calculated a weighted sum of the two binarized tracks (see Table 1 for the weights). Time points at which the weighted track was above the sharp turn threshold multiple were designated as sharp turns, and points below the curved walk multiple were designated as curved walks (S2 Fig). Points with values between these two thresholds were assigned to sharp turns or curved walk based on temporal proximity. If the point was nearest to a stretch (defined as >5 frames) of sharp turn, it was assigned as a sharp turn, otherwise it was assigned as curved walk.

The procedure above–including the various thresholds—was determined empirically. A MATLAB program using the global search algorithm from the Global Optimization Toolbox (Mathworks, Natick, MA Release 2018a) was written to fit the parameters for our segmentation algorithm. We minimized the root mean squared error between the empirical positional tracks and the recreated tracks. The resulting best fit parameter set were as follows:

**Assessment of the contribution of slip and yaw to curvature during sharp turns**

Percent contribution of yaw (p) was calculated as a function of the normalized duration of sharp turns in each normalized time bin (t) as follows:

$$p(t) = \frac{\overline{Y(t)}}{\overline{k(t)}}$$

As the sum of the change in slip angle and the yaw should fully account for the curvature of the trajectory, the percent contribution due to slip angle is simply 1-p.

## Walk-turn-stop-boundary model

Consistent with our analysis in previous studies [37], we first subdivided our behavioral data set into three distinct sets: 1) before first entry (before) into the light-zone, after first entry inside the light-zone (during in), and after first entry outside the light-zone (during out). To generate synthetic flies, we used the empirical distributions from these three distinct sets (S3 Fig and S4 Fig) to create 6-minute long trajectory. Each synthetic run starts at the center with a walk. A duration, speed and curvature is chosen for the walk. Walk terminates in a sharp turn or a stop if the fly is >1.5 mm (half-fly length) away from the boundary. If the fly is within 1.5 mm of the boundary, then the walk terminates in the boundary condition. Stops, sharp turns and boundary conditions all terminate in another walk. This process continues until there are 6 minutes of tracks. First entry is defined by the first time the fly enters the light-zone after the 3-minute mark. How each of the four states are modeled is described below.

## Generation of each of the four states

1. Stops: During a stop, a fly can reorient itself by turning in spot. As such, stop distributions in each scenario were characterized by the joint probability density function (pdf) between duration and total curvature during the stop.

2. Sharp turns: Sharp turns were characterized by the joint pdf between the duration and total curvature of a sharp turn.

3. Curved walks were characterized by the joint pdf between speed, duration, and average instantaneous curvature of a smoothed walk.

4. Boundary distributions were defined by the joint pdf of the duration and total angle of the arc of movement around the boundary.

All pdfs were approximated using a multivariate kernel density estimation function [53]. This algorithm treats the kernel as the transition density of a linear diffusion process and selects bandwidth using a fast and accurate plug-in method.

## Estimation of turn-bias and its implementation in a model

1. Estimation: Sharp turn-in bias was defined as the probability that a fly will turn in a direction that directs them closer to the center of the arena. To empirically calculate the turn-in

bias, we first define a sharp turn as a set of three points such that $p_1$ designates the start of the turn, $p_2$ corresponds to the location of the max curvature, and $p_3$ ended the turn. We defined the movement vectors that connect the three points as:

$$\vec{v}_1 = p_2 - p_1$$

$$\vec{v}_2 = p_3 - p_2$$

We further defined a vector that points radially inwards towards the center of the arena from the sharp turn index ($p_2$) as:

$$\vec{u} = -p_2$$

Next, we calculated the angle the fly trajectory made with $\vec{u}$, the radially inward vector as the fly approached ($\theta_{before}$) the sharp turn and as it left ($\theta_{after}$):

$$\theta_{before} = cos^{-1}\left(\frac{\vec{v}_1 \cdot \vec{u}}{\|\vec{v}_1\|\|\vec{u}\|}\right)$$

$$\theta_{after} = cos^{-1}\left(\frac{\vec{v}_2 \cdot \vec{u}}{\|\vec{v}_2\|\|\vec{u}\|}\right)$$

We note that the fly turned inwards if $\theta_{after} < \theta_{before}$ and outwards if $\theta_{after} > \theta_{before}$. The ratio of the number of inward turning instances over the total number of sharp turns results in the turn-in bias.

2. Implementation. Turn-in bias was implemented in the model as the decision to turn towards the center of the arena. After choosing the sharp turn angle by sampling from the distribution of sharp turn angles, we chose the direction of the turn (the turn can be in two directions) such that the fly points towards the arena center with a probability consistent with the empirical turn bias. Formally, we define a sharp turn choice as a set of three points $\{p_1, p_2, p_3\}$ where $p_1$ initiated the location of the turn, $p_2$ indicated the end location of a left turn and $p_3$ indicated the end location of a right turn. Now we define the movement vectors as:

$$\vec{v}_1 = p_2 - p_1$$

$$\vec{v}_2 = p_3 - p_1$$

We can further define a vector that directs inwards from the sharp turn index ($p_1$) as:

$$\vec{u} = -p_1$$

Next, we calculated the angle the fly trajectory was making with the inward vector as the fly turned left and as it turned right (see definition of biases above). We designate the inward turning choice as the turn that results in a smaller $\theta$. The choice of turn was then decided based upon the empirically derived turn in bias.

## Definition of border choice

When a fly leaves or enters the light zone, they exhibit an increased chance of initiating a turn. We defined the location of the border to be 1.2 cm (0.3 when normalized to arena radius) from the center of the arena. We subdivided our 6-minute trajectories into two scenarios: before (BFE) and after first entry (AFE) based on the first time that a fly enters the light-zone after the light is turned on. For each scenario, we further subdivided the trajectories into inside

tracks (entry of light zone to exit of light zone) and outside tracks (exit to entry). We considered the first two turns and calculated the probability mass function (pmf) as a function of the normalized radial location (r) where these turns occurred for outside ($P_{out}(r)$) and inside ($P_{in}(r)$) tracks separately.

One caveat of this approach is that the pmf will be skewed towards the light border because crossing tracks start at the border. To consider the contribution of this sampling bias, we fit the BFE pmf of sharp turn locations when crossing the arena border to a set of compounding distribution functions.

We employed the compounding distribution functions rather than the raw pmf because the low number of crossing resulted in a noisy empirical distribution. This set of compounding distribution functions were then used to detrend the empirical AFE pmf. The functions were derived as follows:

We begin by making two assumptions:

1. Flies exhibit a uniform radial distribution inside the odor arena before crossing outwards and vice versa for crossing inwards

2. The radial displacement for flies crossing inwards and outwards follow exponential distributions.

Based on these assumptions, the expected distribution of sharp turn locations (T) as a function of normalized radial position (r) depends on the location inside the arena where the walk starts (L) and the expected distribution (D) of radial distance traveled (x) for a given curved walk. Specifically, T can be calculated as a convolution of D and L using the following formula:

$$T(r) = (D * L)(r) = \int_{-\infty}^{\infty} D(x)L(r-x)dx \tag{1}$$

For the outside tracks, let us then define the border location as *b* and thus L represents a uniform distribution ranging from 0 to b. Solving for Eq (1) this we obtain the following:

$$T(r) = \begin{cases} 1 - e^{-\lambda r} & 0 \leq r \leq b \\ e^{-\lambda(r-b)} - e^{-\lambda r} & 1 \geq r > b \\ 0 & o/w \end{cases}$$

For the inside tracks with the same border b, we can define L as a uniform distribution ranging from b to 1. Solving for Eq (1) this we obtain the following:

$$T(r) = \begin{cases} e^{\lambda(r-b)} - e^{\lambda(r-1)} & 0 \leq r \leq b \\ e^{\lambda r} - 1 & 1 \geq r > b \\ 0 & o/w \end{cases}$$

For both outside and inside tracks, we are interested in the tracks that cross the arena boundary and therefore our final detrend function is as follows:

$$T(r) = \begin{cases} e^{\lambda(r-b)} - e^{\lambda(r-1)} & 0 \leq r \leq b \\ e^{-\lambda(r-b)} - e^{-\lambda r} & 1 \geq r > b \\ 0 & o/w \end{cases}$$

We then substract T(r) from the during odor distribution of sharp turns and pass through a rectifier before normalizing to a total probability of 1 for inside and outside tracks separately

as shown below:

$$f_{outside} = \frac{\max(0, P_{out}(r) - T(r))}{\sum \max(0, P_{out}(r) - T(r))}$$

$$f_{inside} = \frac{\max(0, P_{in}(r) - T(r))}{\sum \max(0, P_{in}(r) - T(r))}$$

We then implemented an additional baseline probability to initiate a turn as without it, synthetic flies often moved farther away from the light ring before initiating a turn as compared to empirical flies. A baseline of 0.2 was chosen in order to replicate the turn density profile of empirical flies.

### Implementation of border choice

Border choice was implemented in the model for the first two turns after crossing the light border. As we considered that the border choice would likely be weaker for the second turn than for the first turn, we implemented the border choice as follows:

$$f_{inside}(n) = f_{inside} * P(at\ least\ n\ turns|inside\ track)$$

$$f_{outside}(n) = f_{outside} * P(at\ least\ n\ turns|outside\ track)$$

$$\forall n \in \{1, 2\}$$

### Initialization of synthetic flies

All synthetic flies were initialized to start at the center of a unit circular arena (normalized) centered at (0,0) with an initial heading direction along the positive x-axis (0 degrees). All flies are initialized to select a curved walk as the first state that it enters.

### Selection of synthetic flies

We used the same criteria for including the synthetic flies in our dataset as empirical flies. The flies were included if they reached a distance of at least 1.1x the radius of the light bound outside and 0.9x the radius of the light bound inside in both the before and during scenarios. Furthermore, we selected flies that had a first entry time that is within the 85th percentile of empirical first entry times.

### Comparison of locomotor features

Below, we describe each description of locomotion used in this paper. In each description, we separate out the locomotor tracks into before first entry and after first entry

1. Radial occupancy: The overall probability distribution of the average fly being a radial distance away. A bin size of 0.1 radial units (4 mm) was utilized in generating the histogram distribution.

2. Attraction index: The amount of time a fly spends inside the light ring divided by the total time a fly is in a given scenario.

3. Radial density of turns: This is the the probability mass function of the density of turns. It is calculated as below:

$$f_R(r) = \frac{P_R(r_k)}{\pi(r_{k+1}^2 - r_k^2)}$$

$$f_R(r) = \frac{f_R(r)}{\sum_0^R f_R(r)}$$

Where r is the radial distance away from the center of the arena and k is the bin number.

## Quantification and statistical analysis

The data was analysized using estimation methods to calculate mean, mean differences, and confidence intervals using a MatLab toolbox [54,55]. Scatter plots show individual data points and corresponding error bars show mean and bootstrapped 95% confidence interval (resampled 10000 times, bias-corrected, and accelerated). 95% confidence interval for differences between means were calculated using the same boostrapping methods. P-values were further generated using wilcoxon rank-sum tests and reported in the legends for *pro forma* reporting.

## Supporting information

**S1 Fig. Flies flexibly employ slip, thrust and yaw to turn. A.** Definitions of fly centric kinematic descriptors (thrust, slip and yaw) and world coordinate descriptors (speed and curvature). **B**. Example segment of a fly's trajectory showing that sharp turns do not always display peaks in yaw. Black and red lines show body position of the fly, gray lines show the orientation of the fly, and black dots show head position. Some sharp turns display a smooth change in orientation while others include sharp instantaneous yaw. **C.** Flies flexibly employ slip and yaw to turn. Top: Even when there is a change in yaw during the sharp turn the yaw occurs at different times during the sharp turn. The percent contribution of slip and yaw to sharp turns as a function of the percent progress of completing sharp turns changes with speed. Bottom: A large percentage of sharp turns simply don't show a peak in yaw but this percentage decreases with speed. Blue and red dots represent the time of yaw and curvature peaks. **D.** Flies reported by AY. Katsov and colleagues move at much faster speed possibly due to higher temperatures and because the experiments were performed in the presence of light. Speed distribution for sharp turns and curved walks in the Orco Control (n = 31 flies) and Katsov flies in a large arena (n = 9456 tracks). Data obtained from Dryad (doi.org/10.5061/dryad.854j2). **E.** Characterization of the percent contribution of slip and yaw on sharp turns as a function of the percent progress of completing sharp turns for turns moving at less than 10mm/s (left), between 10 and 20mm/s (middle), and larger than 20mm/s (right).
(TIF)

**S2 Fig. Sharp turns and curved walks can be segmented based on curvature. A.** A sample trajectory after delineation into curved walks and sharp turns using the algorithm in panel B. Gray circle designates arena boundary. **B**. Schematic for delineating sharp turns and curved walks. First the curvature and change in curvature are binarized. Then a weighted sum of the binarized values are calculated and smoothed. Finally, two global thresholds for sharp turns and curved walks are set and values between these thresholds are assigned based on time to nearest sharp turn and curved walk. **C** left: Cumulative probability function for the average curvature of sharp turns (red line) and curved walks (black line). right: The receiver operating characteristic curve for classification by a logistic regression with the average curvature shows

high separability between sharp turns and curved walks.
(TIF)

**S3 Fig. Joint probability distributions for sharp turn and stops.** Stop and turn states for Orco control ($A_1$) and Orco retinal ($A_2$) flies are described by the joint probability density function between curvature and duration. Sharp turns for Orco control ($B_1$) and Orco retinal ($B_2$) flies are described by the joint probability density function between curvature and duration.
(TIF)

**S4 Fig. Joint probability distributions for curved walk and boundary.** Curved walks are described by the joint probability density function between curvature, speed, and duration of sharp turns. Shaded regions represent the 50th, 65th, 80th, and 90th percentile contours for Orco control distributions ($A_1$) and Orco retinal distributions ($A_2$). Arena boundary conditions for Orco control ($B_1$) and Orco retinal ($B_2$) flies are described by the joint probability density function between the central angle of movement and duration.
(TIF)

**S5 Fig. Details of the walk turn stop boundary (WTSB) model.** When reaching the boundary, a synthetic fly selects an angular velocity, duration, and direction of movement. The fly leaves the boundary by orienting towards the center of the arena at an angular offset of +/- 10 degrees. When flies stop, they first stop for a duration before reorienting prior to initiating an directed run. Synthetic flies move with constant velocity and duration during directed runs until either the selected duration expires or they reach the arena boundary. They perform sharp turns by moving in a straight line for half of the duration of the sharp turn before reorienting and then in a straight line for the second half of the sharp turn.
(TIF)

**S6 Fig. Tracks of empirical Orco retinal flies arranged in the increasing order of attraction index.** Green and red lines show tracks prior to and after first entry. Pink region indicates the light region. The outer black circle indicates the arena bound.
(TIF)

**S7 Fig. Synthetic flies based on purely kinematic parameters (WTSB model alone).** Tracks of synthetic Orco retinal flies obtained by the kinematic model arranged in the increasing order of attraction index. Green and red lines show tracks prior to and after first entry. Pink region indicates the light region. The outer black circle indicates the arena bound.
(TIF)

**S8 Fig. Tracks of synthetic Orco retinal flies obtained when the border choice and turn bias are considered in the kinematic model.** Tracks are arranged in the increasing order of attraction index. Green and red lines show tracks prior to and after first entry. Pink region indicates the light region. The outer black circle indicates the arena bound.
(TIF)

**S9 Fig. Comparison of attraction index between Orco retinal synthetic flies obtained by the kinematic model for the before and during light periods and Orco retinal synthetic flies obtained by the final model is shown.**
(TIF)

**S10 Fig. Single antenna Orco flies. A.** Schematic showing a sharp turn and the area covered by the sharp turn. **B**. Left: Cumulative density function for left and right turns of single antenna Orco flies. Left turns cover significantly higher area (two-sample KS test, $p < 0.005$)

despite similar number of turns (n = 484 left turns, n = 456 right turns). Middle: Much of this is contributed by the first two turns each time the flies enter the light ring (two-sample KS test, p<0.001, n = 68 left turns, n = 58 right turns). Right: There was an insignificant difference in the area covered by the first two turns outside the light ring (two-sample KS test, p = 0.366, n = 113 left turns, n = 111 right turns). **C.** Tracks of single antenna Orco flies arranged in the increasing order of attraction index. Green and red lines show tracks prior to and after first entry. Pink region indicates the light region. The outer black circle indicates the arena bound. (TIF)

**S11 Fig. Experimental apparatus. An IR video camera captures the movement of flies in a circular arena.** The arena is comprised of two Plexiglass plates (Ø90 x 1.5 mm high), a spacer (Ø90 mm outside, Ø80 mm inside x 3 mm high), and a black Delrin plate (Ø90 mm outside, Ø25 mm inside X 3 mm high). Drosophila walks in the 3 mm high region surrounded by the spacer. The video camera takes a 6-minute recording of the fly. During the last 3 minutes, a DAQ triggers the LED driver to walks in the 3 mm high region surrounded by the spacer. A condenser lens is placed above the LED to collimate the scattered light beam. The collimated light beam is then passed through a plano-convex lens to create a focused light spot that covers the 25 mm hole in the bottom of the arena. (TIF)

**S1 Video. Sample 10s segment of a fly's trajectory reconstructed from image processing.** The fly is shown as a black oval with a major axis radius of 1.5 mm and minor axis radius of 0.6 mm for visualization. Black lines represent centroid locations during curved walks. Red lines represent centroid locations during sharp turns. Gray arrowheads represent the orientation of the fly. Black dots represent the head location of the fly at corresponding to the corresponding gray arrowhead. For scale, black bars represent 6 mm. (AVI)

## Acknowledgments

We would like to acknowledge the members of Bhandawat lab for discussions, and Nicholas Lent for help with experiments.

## Author Contributions

**Conceptualization:** Liangyu Tao, Vikas Bhandawat.

**Data curation:** Liangyu Tao, Vikas Bhandawat.

**Formal analysis:** Liangyu Tao.

**Funding acquisition:** Vikas Bhandawat.

**Investigation:** Liangyu Tao, Siddhi Ozarkar.

**Methodology:** Liangyu Tao.

**Project administration:** Vikas Bhandawat.

**Resources:** Vikas Bhandawat.

**Software:** Liangyu Tao.

**Supervision:** Vikas Bhandawat.

**Validation:** Liangyu Tao.

**Visualization:** Liangyu Tao, Vikas Bhandawat.

**Writing – original draft:** Liangyu Tao, Vikas Bhandawat.

**Writing – review & editing:** Liangyu Tao, Vikas Bhandawat.

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
