## [Decision Letter · Decision Letter 0]

21 Oct 2019

Dear Dr Bhandawat,

Thank you very much for submitting your manuscript 'Mechanisms underlying attraction to odors in walking Drosophila.' for review by PLOS Computational Biology. Your manuscript has been fully evaluated by the PLOS Computational Biology editorial team and in this case also by independent peer reviewers. The reviewers appreciated the attention to an important problem, but raised some substantial concerns about the manuscript as it currently stands. While your manuscript cannot be accepted in its present form, we are willing to consider a revised version in which the issues raised by the reviewers have been adequately addressed. We cannot, of course, promise publication at that time.

Sincerely,

Bard Ermentrout

Associate Editor

PLOS Computational Biology

Samuel Gershman

Deputy Editor

PLOS Computational Biology

[LINK]

Reviewer's Responses to Questions

**Comments to the Authors:**

Reviewer #1: Tao, Ozarkar, and Bhandawat completed two behavioral experiments using walking Drosophila in a small arena where an optogenetic stimulus provided an appetitive cue in part of the arena. In the first experiment, they compared mutant flies (Orco-Gal4; UAS-Chrimson) to Orco-Gal4 only controls. In the second experiment, they removed one antennae of the fly and observed differences in behavior. In addition, they developed a walk-turn-stop-boundary (WTSB) model that described some aspects of fly behavior based only on the statistics of their movements.

The authors’ decomposition of fly movement into slip, yaw, and thrust was very elegant.

Were the experimental flies male, female or a mix of both?

Please make clear that only retinal-fed flies will express the Chrimson channel.

Were flies housed in the dark prior to experimentation? Fluorescent lights may contain enough red light to activate the Chrimson channel.

It is my understanding that Drosophila and other insects cannot see red light. This seems like a crucial point to make, since flies can exhibit phototactic behavior, and this could be an alternative mechanism to explain their attraction to the optogenetic stimulus.

I’m assuming the wings were removed from the flies prior to the experiment. Please make this point clear.

I’m assuming the Orco-Gal4 gene has been shown to be encoded in ORN’s; and furthermore, that it has been shown to encode something attractive to flies. It would be useful to know what the odorant is (if it is known).

It would be useful for readers to know that fly ORNs are located on the antennae. Therefore, removing an antennae would be reducing a bilateral system to a unilateral one.

In prior work, the same authors concluded that individual flies’ behaviors in a similar task cannot be represented by the behavior of an average fly, but were separable into several distinct clusters based on a 10 state Hidden Markov Model. Compared to the authors’ prior work, the WTSB model seems to be limited in scope with less utility. Perhaps the authors could make some predictions about untested behaviors using the WTSB?

These two sentences in the abstract seem contradictory. “Second, the fly turns more at the border between light-zone and no-light zone and these turns have an inward bias. Surprisingly, there is no increase in turn rate, rather a large decrease in speed that makes it appear that the flies are turning at the border.” How can the fly turn more, but not increase its turn rate? This result would make me double-check my methods for identifying turns.

The authors repeatedly refer to fly “kinematics.” Encoding of kinematics (motion of muscles) vs kinetics (forces necessary to produce movements) is a debate established in motor neuroscience. Not sure you want to use this jargon here, as it applies to fly neurobiology.

In the introduction, the description of “navigational movements” may be better described as “goal directed movements” or “orienting movements towards a stimulus.”

The authors make a point that path integration is not necessary to describe behavior in their models. However, I feel that the experiments are somewhat limited in their ability to disentangle path integration from “kinematic” models of decision-making in the fly as the appetitive stimulus is relatively large compared to the size of the arena. Path integration may come into play when either a) the appetitive stimulus is small compared to the arena size; or b) the arena size is large with multiple available stimuli to explore.

Results “ORN activation alone in the absence of air can mediate robust attraction.” I believe the authors meant to say in the “absence of wind” or in the “absence of air currents.” However, note that in even relatively still air in the laboratory, there will still be some air currents. It may be worth measuring and reporting these with an anemometer, since flies can exhibit anemotaxis.

It is not clear to me how the authors parse out “stops.” I’m assuming it’s when velocity is below some threshold.

Line 203 states that “the synthetic flies are not as attracted to the light-zone as the empirical flies.” Why would they be? The model has no mechanism to respond to rewarding stimuli.

Line 255 A deeper analysis of the effect of removing an antennae seems to be warranted. The flies appear to spin preferentially in one direction. This suggests to me that the flies were using a binaral or stereo-olfaction mechanism for detecting the edge of the stimulus.

Line 266 “The increased turning at the border is due to the decrease in speed as the fly exits the light-zone.” Is this consistent with klinotaxis or not?

Line 292 “It is clear from our data that flies not only select the duration of each walk, but also its speed and curvature.” Please clarify.

Line 342 “Essentially, a circuit that relates decreasing ORN response to decreased walking speed and increased turn angle during sharp turn is all that is necessary to produce robust attraction to a stimulus.” This seems like a key point. Can you propose such a circuit or how it might be implemented by Drosophila neurobiology?

The Hungarian algorithm approach to determining head position seems novel and may warrant further explanation. Otherwise, please provide a citation where this method has been used before.

1-1.3s LOESS smoothing seems like oversmoothing to me. I would be more comfortable if a uniform smoothing was used throughout.

Speed is given as both variables d and s in the methods section.

I am confused as to the difference in Figure 4D and Figure 5G. This appears to be the same data, but the result is different.

Figure 5B, is it noteworthy that the peak of turn density is near the inflection point of the laser intensity?

Supplementary Figure 1 (right panel). Did you analyze data from the Katsov, et al. study? If so, please state where this data is available or if you got it through contact with the authors.

Supplementary Figure 2B. Please include the y-axis and thresholds in the plots of curvature and derivative of curvature.

Supplementary Figure 3. What is the significance of including this figure?

Supplementary Figure 4 (caption). What do the authors mean by the “central angle of movement”?

Supplementary Figure 6-8. Compliments to the authors on displaying all the raw data.

Reviewer #2: Multiple studies have explored how odors are represented in the peripheral olfactory system of Drosophila and how freely-moving flies respond to odor plumes carried by turbulent air flows. In spite of this work, very little is known about the orientation behavior of flies in response to pure olfactory stimulations. Bhandawat and colleagues fill this important gap by using an elegant experimental paradigm. They take advantage of optogenetics to activate all olfactory receptor neurons (ORNs) in the central region of a circular arena. This effectively creates a binary odorant landscape where flies accumulate in the light zone (or odor zone). Using state-of-the-art behavioral analysis, the authors quantify how odor stimulation modifies the kinematic parameters of the fly locomotion.

To ask whether changes in kinematic parameters are sufficient to reproduce the observed behavior of real flies, the authors turn to agent-based modeling. The simulations point out the insufficiency of a purely kinematic model of foraging. Further analysis bring the authors to establish that a crucial component of the fly's navigation algorithm is the ability to reorient toward the source upon exiting the odor zone. Surprisingly, the authors report that reorientation does not involve a bias of turns toward the gradient but that it arises from a deceleration of the fly followed by sharp turning. Given the idiosyncratic nature of the assay (the all-or-none nature of the light landscape and pan-neuronal activation of the peripheral olfactory system), it is unclear whether this conclusion can be generalized to more natural conditions of olfactory stimulations.

Overall, the research is well designed and neatly executed. The manuscript is clearly written (in particular, the introduction abounds with useful references). The conclusions are interesting and of general interest.

Major comments/suggestions:

1. In lines 113-114, the authors make a case that "even the straight walks are curved". This statement is supported by the trajectory segments shown in Fig 2. But it also begs the question of why many trajectories in Supplementary Fig. 7 include long segments that are perfectly linear. These straight segments appear to be inexistent in Supplementary Fig. 6. Could these straight segments in Supp. Fig 7 be due to a tracking artifacts? If not, the authors should look into the behavioral significance of segments devoid of any curvature.

2. In Figs. 4 and 5, the authors use their model to make a convincing case that a non-reorienting model (WTSB) is insufficient to account for the behavior of real flies stimulated by an odor gradient. They go on to show that the introduction of a "border choice" and a "turn-bias" parameter can account for the ability of flies to re-enter the odor zone upon its exit. This is done by training and "\\validating the model on the same dataset. To validate the model further, I would encourage the authors to monitor the behavior of Orco-Gal4;UAS-Chrimson flies in a gradient with a geometry different from the binary landscape - for instance a Gaussian gradient produced by a filter. Testing the behavior in a smoother light gradient might reveal whether flies are capable of biasing their turns toward the source when the stimulus landscape is not binary.

In the paragraph that starts on line 340, the authors argue that reorientation maneuvers based on slowing down followed by (non-biased) sharp turning could account for the navigation of walking flies. Although this conclusion is plausible, it would be strengthened by a validation of the model in more realistic olfactory landscapes than that on which the model was trained.

3. Fig 5B shows that more turns take place inside the odor zone. This point is also evident from trajectories shown in the main figures. Yet, the authors did not establish the ability of their model to reproduce the behavior of flies inside the odor zone. How do flies behave when they experience an all-or-none increase in ORN activity while re-entering the odor zone? Although one might predict that flies move straight as soon as they reenter the odor zone (a surge response that would be compatible with the model of Alvarez-Salvado & Nagel 2018), real flies appear to behave differently. Is the turning dynamics inside the odor zone the same as that outside the odor zone (excluding the interface)?

4. Do flies behave differently upon optogenetic stimulation of all ORNs compared to when stimulated by real odors as was studied by Jung & Bhandawat in 2015?

5. In line 265, the authors refer to behavior "in the absence of light". Do they mean "in the absence of retinal"? Although this difference might look purely semantic, it is associated with a potentially more profound problem. The authors have not ruled out that flies respond to abrupt changes in red light through their visual and/or thermosensory system. Although flies are not supposed to orient to red light, It would be important to exclude that the all-or-none red stimulation does not induce startles. The authors could conduct this control by extending their analysis to a condition where flies do not experience any light change. In behavior devoid of light stimulation, they could apply a behavioral analysis with a "mocked odor zone" and quantify the change in kinematic parameters when flies exit this "mocked zone". This no-light control would nicely complement the no-retinal control.

5. Could the authors take advantage of their orienting model to account for differences in the behavior observed between unilateral and bilateral sensing in Fig. 6A2 and 6A2? How different would the overall reorientation dynamics be if flies were capable of biasing their turns toward the source (Fig. 6C1)? The agent-based model appears to be ideal to address this question. In their discussion of the results observed with flies having unilateral olfactory function, the authors should refer to previous work by Frye (Duistermars & Frye, CB 2009) and Nagel (Alvarez-Salvado & Nagel, eLife 18).

Minor comments/suggestions:

- Abstract: The authors state that they "create a new behavior". Stating that they created a new conditions of olfactory stimulations might be more appropriate.

- The kinematic analysis is based on the average behavior of multiple flies. Recent work by the lab of de Bivort and the authors has highlighted the high degree of variability in olfactory behaviors across individual flies. Does this inter-individual variability partly invalidate the development of a model based on behavioral averages? Whereas the dataset obtained for individual flies might not be sufficient to build and validate agent-based models, this issue should be addressed in the manuscript. Wouldn't the HHMM framework previously proposed by the authors be perfectly suited for this analysis?

- Classifiers are designed to distinguish curved walks from sharps turns. In Supplementary Fig. 2, the performance of these classifiers is evaluated by conducting a ROC analysis. Presumably, the true and false positives were defined based on a ground truth. However, I didn't find in the text a description of what the ground truth corresponds to. This point should be clarified.

- In Fig. 3A, the authors compare real trajectory segments with synthetic trajectory segments simulated with the kinematic model. How was this comparison achieved? Did the authors make the synthetic fly start with the same initial conditions as the real fly to obtain the segments shown in Fig 3A? If so, how did they select the synthetic segments shown in the comparison with the experimental segments? Additional details related to this comparison between experimental and simulated behaviors should be provided in the figure caption and the main text.

- In Fig. 5C1, the blue and black dots are difficult to tell apart.

- In Fig 5D2, the "first-two-turns" and "later-turn" probability function is defined for radial distance inside the odor zone. Does this mean that the turn bias toward the source is calculated for positions inside the odor zone? Wouldn't this be inconsistent with the definition of the inward turn, which assumes the the position is outside the zone?

- Line 296: I didn't understand why each state must necessarily last 1 second. Could this be further explained in the main text?

Typos:

- Line 45: "random walks has been" -> "random walks have been"

- Line 142: To be consistent with the figure labeling and the main text, 2A2 should be written with 2 as a subscript.

- Line 172: The parenthesis including Figure S3 and Figure S4 should be closed.

**Have all data underlying the figures and results presented in the manuscript been provided?**

Reviewer #1: Yes

Reviewer #2: Yes

PLOS authors have the option to publish the peer review history of their article (what does this mean?). If published, this will include your full peer review and any attached files.

Reviewer #1: Yes: Andrew E. Papale

Reviewer #2: No

---

## [Decision Letter · Decision Letter 1]

9 Jan 2020

Dear Dr Bhandawat,

Thank you very much for submitting your manuscript, 'Mechanisms underlying attraction to odors in walking Drosophila.', to PLOS Computational Biology. As with all papers submitted to the journal, yours was fully evaluated by the PLOS Computational Biology editorial team, and in this case, by independent peer reviewers. The reviewers appreciated the attention to an important topic but identified some aspects of the manuscript that should be improved.

We would therefore like to ask you to modify the manuscript according to the review recommendations before we can consider your manuscript for acceptance. Your revisions should address the specific points made by each reviewer and we encourage you to respond to particular issues Please note while forming your response, if your article is accepted, you may have the opportunity to make the peer review history publicly available. The record will include editor decision letters (with reviews) and your responses to reviewer comments. If eligible, we will contact you to opt in or out.raised.

- Supporting Information uploaded as separate files, titled 'Dataset', 'Figure', 'Table', 'Text', 'Protocol', 'Audio', or 'Video'.

We hope to receive your revised manuscript within the next 30 days. If you anticipate any delay in its return, we ask that you let us know the expected resubmission date by email at ploscompbiol@plos.org.

Sincerely,

Bard Ermentrout

Associate Editor

PLOS Computational Biology

Samuel Gershman

Deputy Editor

PLOS Computational Biology

[LINK]

Reviewer's Responses to Questions

**Comments to the Authors:**

Reviewer #1: First sentence of abstract, change “is” to “are”

Incomplete sentence in abstract: “behavior in which Drosophila or fruit flies explore a circular

arena.”

I would rewrite this section in the abstract for clarity. “First, there are large kinematic changes. Second, the fly turns more at the border between light-zone and no-light-zone and these turns have an inward bias. Surprisingly, there is no increase in turn-rate, rather a large decrease in speed that makes it appear that the flies are turning at the border. Similarly, the inward bias of the turns is a result of the increase in turn angle.”

Author summary change “its” to “their” in the sentence “animals have incomplete information about its environment”

In the author summary, please clarify what is meant by “In most laboratory experiments, the information is complete”

In author summary, “we devise a new behavior” should be we devise a new behavioral task or behavioral paradigm.

Typo in author summary “navigate towards to”

Typo in author summary: “First, its speed in the activated region and its turn rate is much lower than it is elsewhere.”

Typo in author summary: “turns becomes”

Typo in introduction: “However, not every animal movement is aimed at reaching a specific destination and are therefore not navigational.”

Typo on line 20: “Random walk model also works well in some isolated cases”

Typo line 25: “therefore exhibit”

Typo line 35: “in laboratory behavioral experiment”

Typo line 45: “describing the effects of stimulus”

Typo line 55: “non-oriented” change to “non orienting”

Typo line 70: “is” to “are”

Typo line 170: “Tracks corresponding to each transition was generated”

Line 180 “fly” to “flies”

Line 193: “modulate” to “modulates”

Line 265: Is it simply because the flies spend more time closer to the border of the stimulus that they are more likely to turn more? And you’re arguing against klinotaxis because the time before first (sharp) turn after leaving the stimulus is not longer in retinal flies? To be fair, you’ve made it pretty difficult to navigate via klinotaxis with the abrupt step-function light stimulus, but off the top of my head I can’t think of a hole in the logic. Does it maybe matter whether you consider the inner “edge” or the outer “edge” to be when the flies “leave the stimulus”?

Line 305: Typo add “this”

Line 331: I would hedge the description of counter turning in moths as an “automatic, internally stored mechanism” that is “released.” It sounds like Kennedy (1983) suggests it might be in his/her review, so maybe just add the phrase, “As suggested by Kennedy” if the authors feel this is indeed an appropriate conclusion from this review. In my opinion, the evidence is far from conclusive however.

Reviewer #2: The authors have essentially addressed the concerns I raised in my previous report. I appreciate the reason why they did not characterize orientation responses to graded stimuli (all-or-none nature of odor plumes). I will not insist on having them include this graded analysis to the present manuscript and, since they are "actively working on these issues", I look forward to seeing their results in a future manuscript. Although I am very supportive of this study and praise its quality, I remain puzzled about the following two points:

1. The existence of long and straight trajectory segments (walks) in the model simulations. I believe I understand the origin of these trajectories - the explanations of the authors are quite clear. But these explanations do not mitigate the fact that the presence of a large number of straight walks from one side of the arena to another points out one aspect of the model that is unrealistic. In case what I am referring to is unclear, I indicated some of the straight walks with a blue arrow in a PDF in attachment. If these straight walks arise from the fact that the average curvature of a trajectory segment is determined prior to the walk, doesn't the discrepancy between simulations and experimental data argue that local curvature should be updated continuously during a walk?

2. I had recommended that the authors look into the ability of their model to reproduce the behavior of flies inside the odor zone. In particular, I suggested analyzing the behavioral dynamics that follows re-entry into the odor zone. The authors answered "Excellent point. We show that the model does replicate the behavior of the flies. To address this critique, we have split Figure 5 into Figures 5 and 6." As explained, the authors split the content of the previous Figure 5 into two figures and added one panel (6B) that does not really address my recommendation (the turn density is not informative about the straightness of the walks). No other changes were apparently made. Am I missing something?

**Have all data underlying the figures and results presented in the manuscript been provided?**

Reviewer #1: Yes

Reviewer #2: Yes

PLOS authors have the option to publish the peer review history of their article (what does this mean?). If published, this will include your full peer review and any attached files.

Reviewer #1: Yes: Andrew E. Papale

Reviewer #2: No

---

## [Decision Letter · Decision Letter 2]

7 Feb 2020

Dear Dr. Bhandawat,

We are pleased to inform you that your manuscript 'Mechanisms underlying attraction to odors in walking Drosophila.' has been provisionally accepted for publication in PLOS Computational Biology.

Before your manuscript can be formally accepted you will need to complete some formatting changes, which you will receive in a follow up email. A member of our team will be in touch within two working days with a set of requests.

Best regards,

Bard Ermentrout

Associate Editor

PLOS Computational Biology

Samuel Gershman

Deputy Editor

PLOS Computational Biology

Reviewer's Responses to Questions

**Comments to the Authors:**

Reviewer #2: I have no additional comments.

**Have all data underlying the figures and results presented in the manuscript been provided?**

Reviewer #2: Yes

PLOS authors have the option to publish the peer review history of their article (what does this mean?). If published, this will include your full peer review and any attached files.

Reviewer #2: No

---

## [Editor Report · Acceptance letter]

11 Mar 2020

PCOMPBIOL-D-19-01609R2 

Mechanisms underlying attraction to odors in walking Drosophila.

Dear Dr Bhandawat,

I am pleased to inform you that your manuscript has been formally accepted for publication in PLOS Computational Biology. Your manuscript is now with our production department and you will be notified of the publication date in due course.

With kind regards,

Laura Mallard
